# Minimizing the number of optimizations for efficient community dynamic flux balance analysis

**James D. Brunner**⦾*, **Nicholas Chia**⦾

Department of Surgery, Center for Individualized Medicine Microbiome Program, Mayo Clinic, Rochester, MN, USA

* brunner.james@mayo.edu

**Data Availability Statement:** The source code is available at https://github.com/jdbrunner/surfin_fba.

**Funding:** This work was supported by funding from the DeWitt and Curtiss Family Foundation

## Abstract

Dynamic flux balance analysis uses a quasi-steady state assumption to calculate an organism's metabolic activity at each time-step of a dynamic simulation, using the well-known technique of flux balance analysis. For microbial communities, this calculation is especially costly and involves solving a linear constrained optimization problem for each member of the community at each time step. However, this is unnecessary and inefficient, as prior solutions can be used to inform future time steps. Here, we show that a basis for the space of internal fluxes can be chosen for each microbe in a community and this basis can be used to simulate forward by solving a relatively inexpensive system of linear equations at most time steps. We can use this solution as long as the resulting metabolic activity remains within the optimization problem's constraints (i.e. the solution to the linear system of equations remains a feasible to the linear program). As the solution becomes infeasible, it first becomes a feasible but degenerate solution to the optimization problem, and we can solve a different but related optimization problem to choose an appropriate basis to continue forward simulation. We demonstrate the efficiency and robustness of our method by comparing with currently used methods on a four species community, and show that our method requires at least 91% fewer optimizations to be solved. For reproducibility, we prototyped the method using Python. Source code is available at https://github.com/jdbrunner/surfin_fba.

## Author summary

The standard methods in the field for dynamic flux balance analysis (FBA) carry a prohibitively high computational cost because it requires solving a linear optimization problem at each time-step. We have developed a novel method for producing solutions to this dynamical system which greatly reduces the number of optimization problems that must be solved. We prove mathematically that we can solve the optimization problem once and simulate the system forward as an ordinary differential equation (ODE) for some time interval, and solutions to this ODE provide solutions to the optimization problem. Eventually, the system reaches an easily check-able condition which implies that another

(JDB), National Cancer Institute grant R01 CA179243 (NC), and the Center for Individualized Medicine, Mayo Clinic (NC and JDB). The funders had no role in study design, data collection and analysis, decision to publish, or preparation of the manuscript.

**Competing interests:** The authors have declared that no competing interests exist.

optimization problem must be solved. We compare our method against typically used methods for dynamic FBA to validate that it provides equivalent solutions while requiring fewer linear-program solutions.

This is a *PLOS Computational Biology* Methods paper.

## Introduction

### Microbial communities and human health

The makeup of microbial communities is often complex, dynamic, and hard to predict. However, microbial community structure has a profound effect on human health and disease [1–7]. These two facts have led to significant interest in mathematical models which can predict relative abundances among microbes in a community. Various dynamical models have been proposed to explain and predict microbial community population dynamics [8–12]. Among these are models which propose that interactions between species are mediated by the metabolites that each species produces and consumes [13, 14], and there is significant evidence that these models perform better than models which depend on direct interaction between species [15, 16].

Recently, advances in genetic sequencing have allowed the creation of genome-scale models (GEMs) that reflect the internal network of cellular metabolism, and can therefore be used to predict metabolite use and production [17–19]. This technique can be extended to microbial community modeling by combining GEMs of different species. There has been significant interest in using GEMs to predict relative populations of stable microbial communities [20–26]. Community metabolic modeling can not only predict relative populations, but also holds the potential to predict and explain the community metabolite yield, which can have a profound effect on health [4]. Furthermore, model repositories such as the online bacterial bioinformatics resource *PATRIC* [27] or the *BiGG model database* [28] make it possible to build community models using information from individual species investigations.

GEMs can be used to predict microbial growth rates as well as metabolite consumption and production rates using a process called *flux balance analysis* (FBA). Because these predictions appear in the form of rates of change, they can be used to define a metabolite mediated dynamical model simply by taking as a vector field the rates of change predicted by FBA. We can therefore combine the techniques of metabolite mediated dynamic modeling and community metabolic modeling to produce dynamic predictions of microbial community population size and metabolite yield. This strategy is called *dynamic FBA* [29–31], and has recently been used to model microbial communities [32–34].

Dynamic FBA, when implemented naïvely, requires a linear optimization problem to be repeatedly solved, and carries a high computational cost for even small communities. Furthermore, *in silico* experiments may need to be repeated many times over various environmental conditions or using various parameter choices in order to make robust conclusions or to accurately fit model parameters. As a result, implementations of dynamic FBA which depend on optimization at every time-step carry a prohibitively high computational cost when used to simulate larger microbial communities. The implementation of dynamic FBA in the popular COBRA toolbox software package [17] is done in this way, and essentially all more efficient

available tools for simulating dynamic FBA fundamentally use an ODE solver approach with optimization at each time-step [24, 31, 35–38]. Dynamic FBA can be improved by taking advantage of the linear structure of the optimization problem which provides a choice of basis for an optimal solution that may be reused at future time-steps [39, 40]. However, the optimizations that are required by this strategy involve solutions with non-unique bases. This means that a basis chosen at random may not provide an optimal solution to the linear program at future time-steps because it provides a solution that is non-optimal or infeasible.

In order to implement dynamic FBA without optimizing at each time step, we use an optimal basic set for the FBA linear optimization problem to create a system of linear equations whose solutions at future time-steps coincide with the solutions to the FBA optimization problem. To solve the problem of non-uniqueness among bases, we prove that there exists a choice of basis that allows forward simulation for a given optimal flux solution and provide a method to choose this basis. Note that this method does not choose among a set of non-unique optimal flux solutions, but instead chooses a basis for a single given optimum. To choose among multiple optimal flux solutions, biological, rather than mathematical, considerations should be used.

In this manuscript, we detail how dynamic FBA can be simulated forward without re-optimization for some time interval, and give a method for doing so. We propose conditions on an optimal basic set for the FBA linear optimization problem which allows for forward simulation, and we prove that such a choice exists. We then detail how to choose this basis set, and finally give examples of simulations which demonstrate the power of our method. For reproducibility, we make a prototype implementation of our method in the Python language available at https://github.com/jdbrunner/surfin_fba.

## Background

### Flux balance analysis

With the advent of genetic sequencing and the resulting genome scale reconstruction of metabolic pathways, methods have been developed to analyze and draw insight from such large scale models [18]. To enable computation of relevant model outcomes, constraint based reconstruction and analysis (COBRA) is used to model steady state fluxes $v_i$ through a microorganism's internal metabolic reactions under physically relevant constraints [18]. One of the most basic COBRA methods, called *flux balance analysis* (FBA) optimizes some combination of reaction fluxes $\Sigma \gamma_i v_i$ which correspond to increased cellular biomass, subject to the constraint that the cell's internal metabolism is at equilibrium:

$$\Gamma \boldsymbol{v} = 0 \tag{1}$$

where $\Gamma$ is the *stoichiometric matrix*, a matrix describing the stoichiometry of the metabolic model.

This optimization is chosen because it reflects the optimization carried out by nature through evolution [18]. The vector $\boldsymbol{\gamma} = (\gamma_1, \gamma_2, \ldots, \gamma_d)$ is an encoding of cellular objectives, reflecting the belief that the cell will be optimized to carry out these objectives. The constraint Eq (1) means that any optimal set of fluxes found by FBA corresponds to a steady state of the classical model of chemical reaction networks [41]. This reflects the assumption that the cell will approach an internal chemical equilibrium.

The optimization is done over a polytope of feasible solutions defined by the inequalities $v_{i,min} \leq v_i \leq v_{i,max}$, or possibly more complicated linear constraints. See Fig 1 for a geometric representation of an example of the type of linear optimization problem that is carried out. By convention, forward and reverse reactions are not separated and so negative flux is allowed. Linear optimization problems like FBA often give rise to an infinite set of optimal flux vectors

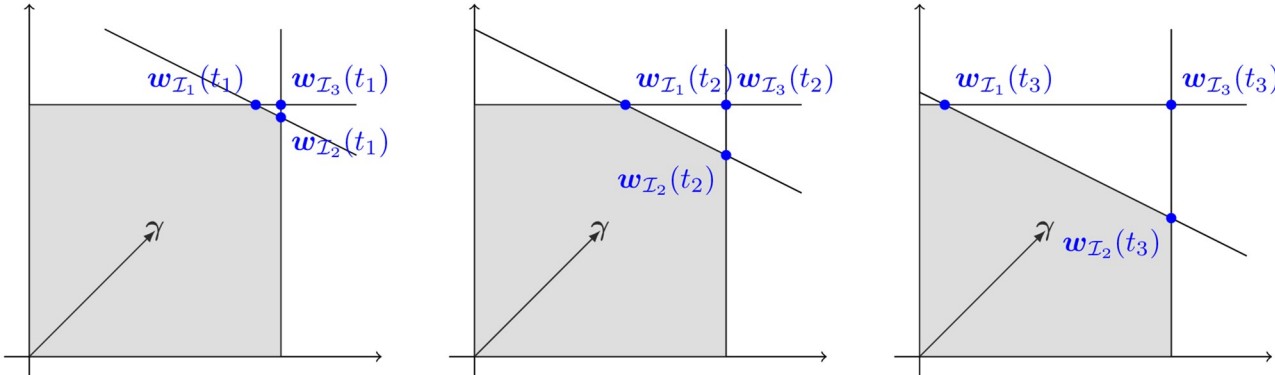

**Fig 1. Geometric representation of Example 1 for $t_3 > t_2 > t_1 > 0$, showing the three options for bases which are equivalent at $t = 0$.** Note that the best choice depends on the function $c(t) = (10, 10, 30 - t)$ and cannot be chosen using the static problem alone. The feasible region of the optimization problem is shown in gray.

$v = (v_1, v_2, \ldots, v_d)$. Geometrically, this set will correspond to some face of the polytope of feasible solutions. To draw conclusions despite this limitation, many methods have been developed to either characterize the set of optimal solutions, as with flux variability analysis (FVA), or enforce more constraints on the network to reduce the size of this set, as with loopless FVA [18].

## Dynamic FBA

FBA provides a rate of increase of biomass which can be interpreted as a growth rate for a cell. Furthermore, a subset of the reactions of a GEM represent metabolite exchange between the cell and its environment. By interpreting constraints on nutrient exchange reactions within the metabolic network as functions of the available external metabolites and fluxes of exchange reactions as metabolite exchange rates between the cell and its environment, the coupled system can be modeled. The simplest way to do this is to use an Euler method, as in [30].

In addition to Euler's method, more sophisticated ODE solvers may be used in the so-called "direct" method of simply recomputing the FBA optimization at every time-step. This can provide better solution accuracy and potentially larger time-steps, but may also require more than one FBA optimization at each time-step. For instance, the Runge-Kutta fourth order method [42] requires four FBA solutions at each time step. Direct methods are implemented in the COBRA toolbox [17] and are the central algorithm in many modern tools, including those of Zhuang et al. [31, 35], Harcombe et al. [36], Zomorrodi et al. [24], Louca and Doebeli [37], and Popp and Centler [38]. Notably, any direct method requires at least one complete recalculation of the network fluxes *at each time-step*.

However, resolving the system at each time step is not necessary, as the solution the optimization problem at some initial time can actually be used to compute future optimal solutions. Höffner et al., [40], used this observation to introduce a variable step-size method for dynamic FBA. In that method a basic index set is chosen by adding biological constraints to the optimization problem hierarchically until a unique optimal flux vector is found. The challenge of such an approach is in choosing the basis for the optimal solution, as the optimal basis is not guaranteed to be unique even for a unique optimal flux solution. In fact, due to the nature of the method of Höffner et al. and of our method, any optimization past the initial solution that must be carried out is guaranteed to have a solution with a non-unique basis. Furthermore, many choices of optimal basis will not provide a solution for future time-steps, so that

choosing among these bases must be done intelligently. Unfortunately, Höffner et al. [40] do not provide a method for choosing among non-unique bases for a single linear program solution.

Our method seeks to solve this problem by choosing a basis which is most likely to remain optimal as simulation proceeds forward from the possibilities provided by an FBA solution. We therefore prioritize reducing the number of times the linear program must be solved, choosing our basis based on the mathematical properties of the system which gives the best chance of providing a solution at future time-steps.

Additionally, a method described as the "dynamic optimization approach" was introduced in Mahadevan et al., [29], however this method is computationally expensive. In particular, the method given in [29] involves optimizing over the entire time-course simulated, and so is formulated as a non-linear program which only needs to be solved once. While this method requires only one optimization, this optimization is itself prohibitively difficult due to the dimensionality of the problem growing with the fineness of time-discretization.

## The dynamic FBA model for communities

We can write a metabolite mediated model for the population dynamics of a community of organisms $x = (x_1, \ldots, x_p)$ on a medium composed of nutrients $y = (y_1, \ldots, y_m)$:

$$\dot{x}_i = g_i(\psi_i(y))x_i \tag{2}$$

$$\dot{y}_j = -\sum_{i=1}^{p}\psi_{ij}(y)x_i \tag{3}$$

where $\psi_i$ is a vector of the fluxes of nutrient exchange reactions for organism $x_i$ as determined by FBA. Using FBA to determine $\psi_i$ is therefore a quasi-steady state assumption on the internal metabolism of the organisms $x_i$ [43–45].

Recall that the basic assumption of flux balance analysis is that, given a matrix $\Gamma_i$ that gives the stoichiometry of the network of reactions in a cell of organism $x_i$ that growth $g_i(y)$ is the maximum determined by solving the following linear program [18]:

$$\left\{ \begin{array}{c} \max{(v_i \cdot \gamma_i)} \\ \Gamma_i v_i = 0 \\ c_i^1 \leq v \leq c_i^2(y) \end{array} \right\} \tag{4}$$

where $c_i^1$ is some vector of lower flux bounds while $c_i^2(y)$ is some vector-valued function of the available metabolites which represents upper flux bounds. The key observation allowing dynamic FBA is that the optimal solution to this problem also determines $\psi_i$ simply by taking $\psi_{ij}$ to be the value of the flux $v_{ij}$ of the appropriate metabolite exchange reaction. For clarity, we will relabel the elements of $v_i$ so that $\psi_{ik} = v_{ij}$ if $v_{ij}$ is the $k^{th}$ exchange flux, and $\phi_{ik} = v_{ij}$ if $v_{ij}$ is the $k^{th}$ internal flux. The objective vector $\gamma_i$ indicates which reactions within the cell contribute directly to cellular biomass, and so is non-zero only in elements corresponding to internal fluxes. We can therefore rewrite this vector to include only elements corresponding to internal fluxes, so that the objective of the optimization is to maximize $\gamma_i \cdot \phi_i$.

The stoichiometry of metabolite exchange reactions is represented by standard basis vectors [18]. Therefore, we can partition $\Gamma_i$ as

$$\Gamma_i = \begin{bmatrix} I & -\Gamma_i^* \\ 0 & \Gamma_i^\dagger \end{bmatrix} \tag{5}$$

where $I$ is the identity matrix of appropriate size, and $\Gamma_i^*$ and $\Gamma_i^\dagger$ contain the stoichiometry of the internal reactions [18, 46, 47]. Making this change in notation allows us to see that the optimization problem of flux balance analysis is essentially internal to the cell, with external reactions providing constraints.

We can see from Eq (5) that $\ker(\Gamma_i)$ is isomorphic to $\ker(\Gamma_i^\dagger)$, and so we can maximize over this kernel. Then, the exchange reaction fluxes are determined by the internal fluxes according to the linear mapping $\psi_i = \Gamma_i^* \phi_i$. The maximization of FBA becomes a maximization problem over the internal fluxes. We rewrite Eq (4) using Eq (5) and combine with Eqs (2) and (3) to form the differential algebraic system

$$\frac{dx_i}{dt} = x_i(\gamma_i \cdot \phi_i) \tag{6}$$

$$\frac{d\mathbf{y}}{dt} = -\sum_i x_i \Gamma_i^* \phi_i \tag{7}$$

$$\left\{ \begin{array}{c} \max\left(\phi_i \cdot \gamma_i\right) \\ \Gamma_i^\dagger \phi_i = 0 \\ c_i^1 \leq \begin{bmatrix} \Gamma_i^* \\ I \end{bmatrix} \phi_i \leq c_i^2(\mathbf{y}) \end{array} \right\} \tag{8}$$

where each $\phi_i$ is determined by the optimization Eq (8), all carried out separately. Note that this is a metabolite mediated model of community growth as defined in [15]. That is, the coupling of the growth of the separate microbes is due to the shared pool of metabolites $\mathbf{y}$. Each separate optimization which determines $\phi_i$ at a single time-step depends on $\mathbf{y}$, and each $\phi_i$ determines some change in $\mathbf{y}$. Furthermore, each optimization is carried out in a manner that depends only the status of the metabolite pool and is independent from the optimizations of other organisms. There is therefore no shared "community objective". Instead, each organism optimizes according to only its own internal objective.

We write, for full generality, upper and lower dynamic bounds on internal and exchange reactions, and assume that each function $c_{ij}(\mathbf{y}) \in C^\infty$. We let

$$A_i = \left[ (\Gamma_i^*)^T, -(\Gamma_i^*)^T, I, -I, \right]^T \tag{9}$$

so that we can rewrite the optimization problem Eq (8) as

$$\left\{ \begin{array}{c} \max\left(\phi_i \cdot \gamma_i\right) \\ A_i \phi_i \leq c_i(\mathbf{y}, t) \\ \Gamma_i^\dagger \phi_i = \mathbf{0} \end{array} \right\} \tag{10}$$

for ease of notation.

We now hope to select a basic index set $\mathcal{I}_i$ for Eq (10) for each organism $x_i$ so that each $\boldsymbol{\phi}_i(t)$ is a solution to the resulting linear system of equations.

## Methods

### Linear optimization preliminaries

In this manuscript, we will rewrite the FBA optimization problem in the form

$$\left\{ \begin{array}{c} \max{(\boldsymbol{\phi} \cdot \boldsymbol{\gamma})} \\[6pt] A\boldsymbol{\phi} \leq \boldsymbol{c} \\[6pt] \Gamma^{\dagger}\boldsymbol{\phi} = 0 \end{array} \right\} \tag{11}$$

where the matrices $A$ and $\Gamma^{\dagger}$ are derived from the stoichiometric matrix and flux constraints. Such a problem is often referred to as a *linear program* (LP). We now recall some well known results from the study of linear programming (see, for example [40, 48]).

First, we note that Eq (11) can be rewritten in the so-called *standard form* with the addition of *slack variables* $\boldsymbol{s} = (s_1, \ldots, s_n)$ which represent the distance each of the $n$ constraints is from its bound as follows:

$$\left\{ \begin{array}{c} \max{(\tilde{\boldsymbol{\phi}} \cdot \tilde{\boldsymbol{\gamma}})} \\[8pt] \begin{bmatrix} \tilde{A} & I \end{bmatrix} \begin{bmatrix} \tilde{\boldsymbol{\phi}} \\ \boldsymbol{s} \end{bmatrix} = \boldsymbol{c} \\[8pt] \tilde{\phi}_i \geq 0, s_i \geq 0 \end{array} \right\}. \tag{12}$$

Standard form requires that we rewrite $\boldsymbol{\phi}_i = \boldsymbol{\phi}_i^+ - \boldsymbol{\phi}_i^-$ and then define $\tilde{\boldsymbol{\phi}} = (\phi_1^+, \phi_2^+, ..., \phi_d^+, \phi_1^-, \phi_2^-, ..., \phi_d^-)$ so that we require non-negativity of each variable, and the matrix $\tilde{A} = [A\ B]$, $B = -A$. We rewrite the problem in this form to make use of established results, and for ease of notation will write $\boldsymbol{\phi}$ instead of $\tilde{\phi}$ when it is clear which form of the problem we are discussing.

We will make use of the well-known result that there exists an *optimal basis* or *basic set* for a bounded linear program [49]. To state this result, we first define the notation $B_{\mathcal{J}}$ to be the matrix with columns of $[\tilde{A}\ I]$ corresponding to some index set $\{k_1, k_2, ..., k_n\} = \mathcal{J}$, and if $B_{\mathcal{J}}$ is invertible we define the notation $\boldsymbol{w}_{\mathcal{J}}(\boldsymbol{a})$ so that

$$(\boldsymbol{w}_{\mathcal{J}}(\boldsymbol{a}))_l = \begin{cases} (B_{\mathcal{I}}^{-1}\boldsymbol{a})_j & l = k_j \in \mathcal{J} \\[6pt] 0 & l \notin \mathcal{J} \end{cases} \tag{13}$$

for any $\boldsymbol{a} \in \mathbb{R}^n$. We may now define a *basic optimal solution* and corresponding *basic index set*.

**Definition 1** *A* basic optimal solution *to a linear program is an optimal solution along with some index set* $\{k_1, k_2, ..., k_n\} = \mathcal{I}$ *such that* $\boldsymbol{w} = \boldsymbol{w}_{\mathcal{I}}(\boldsymbol{c})$, *where* $\boldsymbol{c}$ *is the vector of constraints as in* Eq (12). *The variables* $\{\boldsymbol{w}_i | i \in \mathcal{I}\}$ *are referred to as* basic variables, *and the index set* $\mathcal{I}$ *is referred to as the* basic index set.

Finally, if there exists a bounded, optimal solution to Eq (12), then there exists a basic optimal solution and corresponding basic index set.

For a given basic optimal solution vector $\boldsymbol{w}$, there may be more than one basic index set $\mathcal{I}$ such that $\boldsymbol{w} = \boldsymbol{w}_{\mathcal{I}}(\boldsymbol{b})$. Such a solution is called *degenerate*. Clearly a necessary condition for

such non-uniqueness is that there exists some $k \in \mathcal{I}$ such that $w_k = 0$. This is also a sufficient condition as long as there is some column of $[\tilde{A}\ I]$ which is not in the column space of $B_{\mathcal{I}} \setminus \{k\}$.

## Forward simulation without re-solving

Consider again Eq (10), the linear program that must be solved at each time point of the dynamical system for each microbial population. Information from prior solutions can inform future time-steps as long as the region of feasible solutions has not qualitatively changed. Thus, we may only need to solve the optimization problem a few times over the course of a simulation. The key observation making this possible is that the simplex method of solving a linear program provides an optimal basis for the solution. We may often re-use this basis for future time-steps within some time interval, and therefore find optimal solutions without re-solving the linear program.

In order to do this, we need to find a form of the solution which may be evolved in time. Thus, we turn the system of linear inequalities given in the linear program into a system of linear equations. Then, if this system has a unique solution we have reduced the task to solving a system of equations rather than optimizing over a system of inequalities. We can find such a system of equations by solving the linear program once, and using this solution to create a system of equations whose solution provides the optimal flux $\phi_i$ using a basic index set. We then use this same system to simulate forward without the need to re-solve the optimization problem until the solution to the system of equations is no longer a feasible solution to the linear program.

First, the linear program Eq (10) is transformed into standard form (Eq (12)). Then, a basic optimal solution is found with corresponding basic index set $\mathcal{I}_i$. The dynamical system Eqs (6), (7) and (10) can then be evolved in time using Eq (13). This evolution is accurate until some $w_{ij}$ becomes negative (meaning that the solution is no longer a feasible solution to the linear program). At this point, a new basis must be chosen. That is, until $\boldsymbol{w}_{\mathcal{I}_i}(\boldsymbol{c}(t))$ becomes infeasible, we let $(\phi_{j_1}(\boldsymbol{c}_i(t)), ..., \phi_{j_m}(\boldsymbol{c}_i(t)), s_1(\boldsymbol{c}_i(t)), ..., s_n(\boldsymbol{c}_i(t))) = \boldsymbol{w}_{\mathcal{I}_i}(\boldsymbol{c}_i(t))$ and replace Eqs (6), (7) and (10) with

$$\frac{dx_i}{dt} = x_i(\boldsymbol{\gamma}_i \cdot \boldsymbol{\phi}_i(\boldsymbol{c}_i(t))) \tag{14}$$

$$\frac{d\boldsymbol{y}}{dt} = -\sum_i x_i \Gamma_i^* \boldsymbol{\phi}_i(\boldsymbol{c}_i(t)) \tag{15}$$

One major difficulty in this technique is that a unique $\boldsymbol{w}_i$ does not guarantee a unique basis set $\mathcal{I}_i$. If we have some $(w_{\mathcal{I}_i})_j = 0$ for $j \in \mathcal{I}_i$, then there exists some alternate set $\hat{\mathcal{I}}_i$ such that $\boldsymbol{w}_{\hat{\mathcal{I}}_i} = \boldsymbol{w}_{\mathcal{I}_i}$. Such a solution $\boldsymbol{w}_{\mathcal{I}_i}$ is called *degenerate*. In a static implementation of a linear program, the choice of basis of a degenerate solution is not important, as one is interested in the optimal vector and optimal value. However, as we will demonstrate with 1, the choice of basis of a degenerate solution is important in a dynamic problem. In fact, if the system given in Eqs (14) and (15) is evolved forward until $\boldsymbol{w}_{\mathcal{I}_i}(\boldsymbol{c}_i(t))$ becomes infeasible, the time at which the system becomes infeasible is the time at which we have some $(w_{\mathcal{I}_i})_j = 0$ for $j \in \mathcal{I}_i$. Thus, we need to resolve Eq (10) whenever $\boldsymbol{w}_{\mathcal{I}_i}(\boldsymbol{c}_i(t))$ becomes degenerate, which will be the final time-point at which the $\boldsymbol{w}_{\mathcal{I}_i}(\boldsymbol{c}_i(t))$ is feasible.

**Example 1** *Consider the dynamic linear program*

$$
\left\{
\begin{array}{c}
\max\left((1,1)\cdot \boldsymbol{v}\right)\\[4pt]
\begin{bmatrix} 1 & 0 \\ 0 & 1 \\ 1 & 2 \end{bmatrix} \boldsymbol{v} \le \begin{bmatrix} 10 \\ 10 \\ 30 - t \end{bmatrix}\\[4pt]
v_i \ge 0
\end{array}
\right\}
\tag{16}
$$

*In standard form at t = 0, this linear program becomes*

$$
\left\{
\begin{array}{c}
\max\left((1,1)\cdot \boldsymbol{v}\right)\\[4pt]
\begin{bmatrix} 1 & 0 & 1 & 0 & 0 \\ 0 & 1 & 0 & 1 & 0 \\ 1 & 2 & 0 & 0 & 1 \end{bmatrix} \begin{bmatrix} \boldsymbol{v} \\ \boldsymbol{s} \end{bmatrix} = \begin{bmatrix} 10 \\ 10 \\ 30 \end{bmatrix}\\[4pt]
v_i, s_i \ge 0
\end{array}
\right\}
\tag{17}
$$

*which has the unique solution $\boldsymbol{w} = (10, 10, 0, 0, 0)$. There are three choices of basic index sets: $\mathcal{I}_1 = \{1,2,3\}, \mathcal{I}_2 = \{1,2,4\}$, and $\mathcal{I}_3 = \{1,2,5\}$. The resulting bases are*

$$
B_{\mathcal{I}_1} = \begin{bmatrix} 1 & 0 & 1 \\ 0 & 1 & 0 \\ 1 & 2 & 0 \end{bmatrix} \quad B_{\mathcal{I}_2} = \begin{bmatrix} 1 & 0 & 0 \\ 0 & 1 & 1 \\ 1 & 2 & 0 \end{bmatrix} \quad B_{\mathcal{I}_3} = \begin{bmatrix} 1 & 0 & 0 \\ 0 & 1 & 0 \\ 1 & 2 & 1 \end{bmatrix}
$$

*Computing Eq (13) at $t > 0$ for each, we have that $B_{\mathcal{I}_1}$ yields $\boldsymbol{w}_{\mathcal{I}_1}(\boldsymbol{c}(t)) = (10 - t, 10, t, 0, 0)$, $B_{\mathcal{I}_2}$ yields $\boldsymbol{w}_{\mathcal{I}_2}(\boldsymbol{c}(t)) = (10, 10 - {}^t/_2, 0, {}^t/_2, 0)$, and $B_{\mathcal{I}_3}$ yields $\boldsymbol{w}_{\mathcal{I}_3}(\boldsymbol{c}(t)) = (10, 10, 0, 0, -t)$, shown in Fig 1 for $t > 0$. Thus, only $\boldsymbol{w}_{\mathcal{I}_2}(\boldsymbol{c}(t))$ solves the dynamic problem because $\boldsymbol{w}_{\mathcal{I}_1}(\boldsymbol{c}(t))$ is not optimal and $\boldsymbol{w}_{\mathcal{I}_3}(\boldsymbol{c}(t))$ is not feasible for $t > 0$. We may follow $\boldsymbol{w}_{\mathcal{I}_2}(\boldsymbol{c}(t))$ and be insured of remaining at an optimal solution to the linear program until $t = 20 + \varepsilon$, at which point $\boldsymbol{w}_{\mathcal{I}_2} = (10, -\varepsilon/2, 0, 10, 0)$, which is not a feasible solution to the linear program. At time $t = 20$, a re-optimization is required to choose a new basis.*

*Notice that the correct choice of basis fundamentally depends on the time-varying bound function $\boldsymbol{c}(t) = (10, 10, 30 - t)$. To see this, consider other possible time-varying bounds $\boldsymbol{c}(t)$ which have $\boldsymbol{c}(0) = (10, 10, 30)$. For example, if $\boldsymbol{c}(t) = (10 - t, 10 - t, 30)$, then only $B_{\mathcal{I}_3}$ would give the correct $\boldsymbol{w}(\boldsymbol{c}(t))$ for $t > 0$.*

## A basis for the flux vector

We now provide a method to choose a basis $\mathcal{I}_i$ for each organism $x_i$ in the case of a degenerate solution. Consider an optimal solution $\boldsymbol{w}_i$ to the linear program Eq (12). To simulate forward

according to Eqs (14) and (15), we need for each organism $x_i$ a basic index set $\mathcal{I}_i$ such that

$$
\left\{
\begin{array}{c}
\dot{\boldsymbol{w}}_i = \boldsymbol{w}_{\mathcal{I}_i}\left(\frac{d}{dt}\boldsymbol{c}_i\right) \\[2mm]
\begin{bmatrix} \tilde{A} & I \end{bmatrix}\dot{\boldsymbol{w}} = \frac{d}{dt}\boldsymbol{c}_i \\[2mm]
(\boldsymbol{w}_{\mathcal{I}_i})_j = 0 \Rightarrow \dot{w}_{ij} \geq 0
\end{array}
\right\}
\tag{18}
$$

so that the solution remains feasible, and furthermore that $\dot{\boldsymbol{w}}_i$ is optimal over the possible choice of basic index sets for $\boldsymbol{w}_i$. This is obviously a necessary condition for forward simulation within some non-empty time interval, and can be made sufficient (although no longer necessary) by making the inequality $(\boldsymbol{w}_{\mathcal{I}_i})_j = 0 \Rightarrow \dot{w}_{ij} \geq 0$ strict. We use the relaxed condition for more practical applicability.

In order to develop a method based on the above observation (i.e., Eq (18)), we must know that Eq (12) has such a solution. We therefore require the following lemma, which is proved in S1 Text:

**Lemma 1** *For a linear program with the form given in* Eq (12) *with a basic optimal solution* $\boldsymbol{w}$, *there exists a basic index set* $\mathcal{I}$ *such that* Eq (18) *holds and* $\dot{\boldsymbol{w}}$ *is optimal over the possible choice of basic index sets for* $\boldsymbol{w}$.

If Eq (12) has only a non-degenerate solution, the unique basis will satisfy this requirement. The challenge remains to choose from among the possible bases of a degenerate solution.

To do this, we form a second linear program analogous to Eq (18) in the following way. We first find all constraints $\boldsymbol{a}_{ij}$ (i.e. rows of $A_i$ or $\Gamma_i^\dagger$) such that $\boldsymbol{a}_{ij} \cdot \boldsymbol{\phi}_i = c_{ij}(t)$, calling this set $\mathcal{S}_i$. Note that this set contains all the rows of $\Gamma_i^\dagger$, for which we regard $c_{ij}(t) = 0$ for all $t > 0$. Note that if the solution given is a basic optimal solution, the rank of the matrix whose rows are $\boldsymbol{a}_{ij}$ for $\boldsymbol{a}_{ij} \in \mathcal{S}_i$ is $d$, where again $d$ is the number of internal fluxes. This is true because we include constraints of the type $a < \phi_{ij} < b$ as rows of $A_i$.

Then, we solve the linear program

$$
\left\{
\begin{array}{c}
\max\left(\dot{\boldsymbol{w}}_i \cdot \gamma_i\right) \\[2mm]
\boldsymbol{a}_j \cdot \dot{\boldsymbol{\phi}}_i \leq \frac{dc_{ij}}{dt}, \quad \boldsymbol{a}_j \in \mathcal{S}_i
\end{array}
\right\}
\tag{19}
$$

We may then use any basis $B_{\mathcal{I}}^i$ which solves Eq (19) as long as it has exactly $d$ non-basic slack variables. Lemma 1 tells us that such a choice exists, although it may be necessary to manually pivot non-slack variables into the basis set given by the numerical solver. In testing the algorithm, this was necessary when using IBM ILOG CPLEX Optimization Studio to solve, but not when using The Gurobi Optimizer. Note that we do not need the entire basis $B_{\mathcal{I}}^i$, but instead only need the $d \times d$ submatrix formed by rows of $A_i$ or $\Gamma_i^\dagger$ which correspond to non-basic slack variables in the solution to Eq (19). These appear as rows $(\boldsymbol{a}_i, \boldsymbol{0})$ in $B_{\mathcal{I}}^i$, and so this sub-matrix uniquely determines $\boldsymbol{\phi}_i$. We call this smaller matrix $B_i$, and label the set of row indices as $\mathcal{J}$.

The chosen basis $\mathcal{J}$ and corresponding constraints are used to simulate forward until that particular solution becomes infeasible. At that time, we have an optimal solution to Eq (10) simply by continuity. We therefore do not need to resolve Eq (10) but instead re-form and solve Eq (19).

## Pseudo-Code of the method

Below, we present as pseudo-code an outline of the method. A practical implication may need to adaptively adjust the time-step $\Delta t$ to insure that no resource is artificially over-depleted past 0.

**Algorithm 1**: Dynamic FBA algorithm following Lemma 1. Note that for numerical stability and speed, we may store the matrices $Q_i$, $R_i$ such that $Q_i R_i = B_i$ is the QR-factorization of $B_i$ rather than either storing $B_i^{-1}$ or solving completely during each time step of numerical integration.

```
Input:  Final time T, initial microbial biomasses xᵢ(0), initial nutri-
        ent concentrations yⱼ(0), maximum inflow rates of nutrients αᵢ,
        stoichiometric matrices Γᵢ
OutPut: Timecourse simulation of biomass and nutrient concentrations
1 for each microbial population i do
2    Set wᵢ(0) to be solution to Eq (13) which lies on a vertex of the
     feasible polytope.;
3    Solve Eq (21) to find initial basis Bᵢ
4 end
5 while t < T do
6    Integrate Eqs (14) and (15) from t to t + Δt with φᵢ = Bᵢ⁻¹c_𝒥(y(t),t);
7    if Bᵢ⁻¹c_𝒥(y(t + Δt), t + Δt) is not a feasible solution then
8       reset xᵢ = xᵢ(t), yⱼ = yⱼ(t);
9       Solve Eq (21) to find new basis Bᵢ, with additional constraints
        representing bounds violated by Bᵢ⁻¹c_𝒥(y(t),t).
10   end
11 end
```

## Results

### Number of optimizations

We can compare the efficiency of Algorithm 1 with modern dynamic FBA methods by counting the number of times a large linear program must be carried out over the course of a simulation. At their core, state-of-art dynamic FBA tools such as *d-OptCom* [24] and *COMETS* [36] employ the direct method of calling an ODE-solving method with the linear program set as the right-hand-side. In the case of Euler's method, the resulting ODE can be integrated by hand between time-steps. This last strategy is often referred to as the "static optimization approach" [40].

We compared simulation of various combinations of the organisms *Escherichia coli str. K-12 substr. MG1655* (model iJR904), *Saccharomyces cerevisiae S288C* (model iND705), *Pseudomonas putida KT2440* (model iJN746) and *Mycobacterium tuberculosis H37Rv* (model iEK1008), using models from the BiGG database [28] (see S2 Table for details). We counted the optimizations required for our model, as well as for direct methods using the numerical ODE solvers *vode*, *zvode*, *lsoda*, *dopri5*, and *dop853* from the SciPy library. All of these numerical ODE solvers use adaptive step sizes for accuracy and stability, and so represent optimized choices of time-steps. Additionally, we compared the method of Höffner et al. as implemented in the MatLab package *DFBAlab* [39]. The number of re-optimizations required for each simulation is shown in Table 1 and the time-point of each re-optimization that was carried out is shown in Fig 2.

For our method and the direct method, we allowed exchange of every metabolite detailed in S1 Table with initial metabolite concentrations given by that same file, and with initial biomass of 0.3 for each species. The file sim_comm.py in the supplementary repository https://github.com/jdbrunner/surfin_fba contains complete simulation set-up.

**Table 1. Number of realizations required to simulate to time $t = 5$ with no cell death or metabolite flow, using M9 minimal medium.** *Simulation failed at $t = 3.034277$.

| Model Combination | Solution Method | | | | | | |
|---|---|---|---|---|---|---|---|
| | **Algorithm 1** | **Höffner** | **vode** | **zvode** | **lsoda** | **dopri5** | **dop853** |
| iJR904 | 7 | 1 | 62 | 62 | 116 | 3313 | 6228 |
| iND750 | 4 | 1 | 91 | 91 | 85 | 3508 | 6514 |
| iJN746 | 4 | 13 | 166 | 167 | 376 | 1176 | 2249 |
| iEK1008 | 4 | 4 | 120 | 120 | 208 | 2768 | 5148 |
| iJR904 + iND750 | 4 | 24 | 240 | 211 | 346 | 5586 | 10469 |
| iJR904 + iJN746 | 30 | 479 | 420 | 420 | 744 | 2695 | 5579 |
| iJR904 + iEK1008 | 20 | 136 | 216 | 216 | 454 | 3385 | 6411 |
| iND750 + iEK1008 | 8 | 32 | 311 | 311 | 509 | 5284 | 9888 |
| iJR904 + iND750 + iEK1008 | 18 | 32* | 451 | 451 | 1282 | 6225 | 11961 |
| iJR904 + iND750 + iJN746 + iEK1008 | 56 | 672 | 1122 | 1122 | 2242 | 6837 | 13529 |

To compare with the method of Höffner et al. [40], we use the newly available Python package from the research group of Dr. David Tourigny titled *dynamic-fba* [50] for single organisms. This package allows simulation without secondary optimizations, as our does, and so is more similar to our prototype tool for comparison. Unfortunately, this package is currently only able to simulate single organisms at the time of publishing. For microbial communities, we can compare with the MatLab package DFBAlab [39] which requires all dynamics variables to be optimized in a secondary optimization. For simulations with DFBAlab, we use only the low-concentration metabolites D-glucose, oxygen, and cob(I)alamin from the M9 medium detailed in S1 Table as dynamically varying metabolites. It is worth noting that these are the most favorable conditions we could find for the method of Höffner [39, 40] et al. which are still biologically equivalent to our other simulations.

## Error estimation

Our method provides much less theoretical error in dynamic FBA solutions than traditional methods. In fact, Algorithm 1 implies that a simulation of a microbial community can be

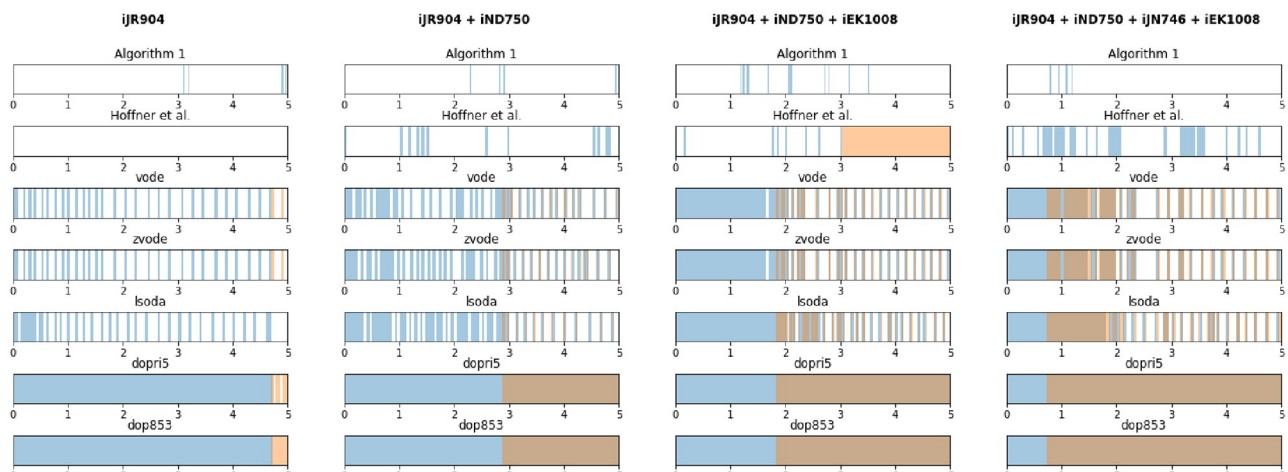

**Fig 2. Time-points of re-optimizations required in simulations using the proposed method, the method of Höffner et al. [40] and various direct methods, shown in blue.** Shown in orange are times at which the direct method solver encountered an infeasible linear program due to numerical error.

divided into time intervals on which the algorithm is exact. Of course, this assumes that the linear ODE solved in these intervals is solved exactly rather than numerically.

Precisely, there exits some sequence $t_0 = 0 < t_1 < \cdots < t_{n-1} < t_n = T$ such that if we know the optimal flux vectors $w_i(t_l)$ at time $t_l$, then Lemma 1 implies the existence of a set of invertible matrices $B_i^l$ such that solutions to Eqs (14) and (15) are solutions to Eqs (6), (7) and (10) for $t \in [t_l, t_{l+1}]$. Therefore, if we are able to identify the $t_l$ exactly, then Algorithm 1 provides exact solutions to the dynamic FBA problem Eqs (6), (7) and (10). Of course, numerical limitations imply that we will not re-optimize precisely at each $t_l$, and so we must investigate the impact of this error. However, once re-optimization is done, the method is again exact. The result is that we have no local truncation error for any time step taken between re-optimization after $t_l$ and the interval endpoint $t_{l+1}$, except for error due to numerical integration. In comparison, direct methods provide some integration error at every time step. This error depends on the integration strategy used, and so for example the Euler's method based static optimization approach carries first order local truncation error at each time step. This can easily lead to ODE overshoot and infeasible linear programs at future time-steps.

Assume that $t_{l-1}$ is known exactly, and $N$ is such that

$$t^1 = t_{l-1} + (N-1)\Delta t \leq t_l < t_{l-1} + N\Delta t = t^2,$$

so that there is some possible error in the interval $[t^1, t^2]$. We can estimate the accumulated error in this time interval using a power series expansion. Let $x(t)$, $y(t)$ be solutions to Eqs (6), (7) and (10) and $\tilde{x}, \tilde{y}$ be solutions given by Algorithm 1 for $t \in [t^1, t^2]$. Furthermore, let $B_i^{l-1}$ be the invertible matrices derived by solving Eq (10) at $t_{l-1}$ and $B_i^l$ those derived by solving at $t_l$. Then, $x(t^1) = \tilde{x}(t^1)$ and $y(t^1) = \tilde{y}(t^1)$. For each $x_i$ we expand, assuming some regularity of the functions $c(y)$,

$$x_i(t^2) - \tilde{x}_i(t^2) = (\Delta t) x_i(t_1)(\gamma_i \cdot ((B_i^{l-1})^{-1} - (B_i^{l-1})^{-1})\hat{c}_i(y(t^1))) + o(\Delta t) \tag{20}$$

and see that this method gives first order local error in time steps that require a re-optimization.

The local error, while first order, only appears at time steps in which a re-optimization occurred, and so global error will scale with the number of necessary re-optimizations. This is in contrast with the classical use of Euler's method, which gives first order local error at every time-step, or any other direct ODE method, whose error is dependent on the solver used.

We may compare the solutions provided by direct methods with those provided by the method presented in Algorithm 1 and by the method of Höffner et al. [40]. The root-sum-square ($l_2$) difference in results are shown in Table 2, and example simulations are shown in Fig 3. As we argue above, direct methods are less accurate in theory that the algorithm presented in Algorithm 1. Furthermore, direct simulations routinely failed to simulate to time $t = 5$ without encountering an infeasible linear program. This infeasibility is the result of numerical error accumulating throughout the simulation. The comparisons in Table 2 can be summarized by three distinct characteristics. First, in the case of *S.cerevisiae*, the direct methods agree well with the newly presented method. Secondly, in the case of *E.coli* and *M.*

**Table 2. $l_2$ difference in solutions to single-organism simulations between direct methods and the method presented in Algorithm 1.**

|  | vode | zvode | lsoda | dopri5 | dop853 | Hoffner et al. |
|---|---|---|---|---|---|---|
| E.coli | 5.09933 | 5.09933 | 4.61467 | 5.09928 | 5.09928 | 4.68578 |
| M.tuberculosis | 1.45401 | 1.45401 | 1.45417 | 1.45415 | 1.45415 | 2.48691 |
| S.cerevisiae | 0.00426 | 0.00426 | 0.00430 | 0.00429 | 0.00429 | 3.06105 |
| P.putida | 15.29177 | 15.29177 | 0.07080 | 15.23826 | 15.26221 | 4.78751 |

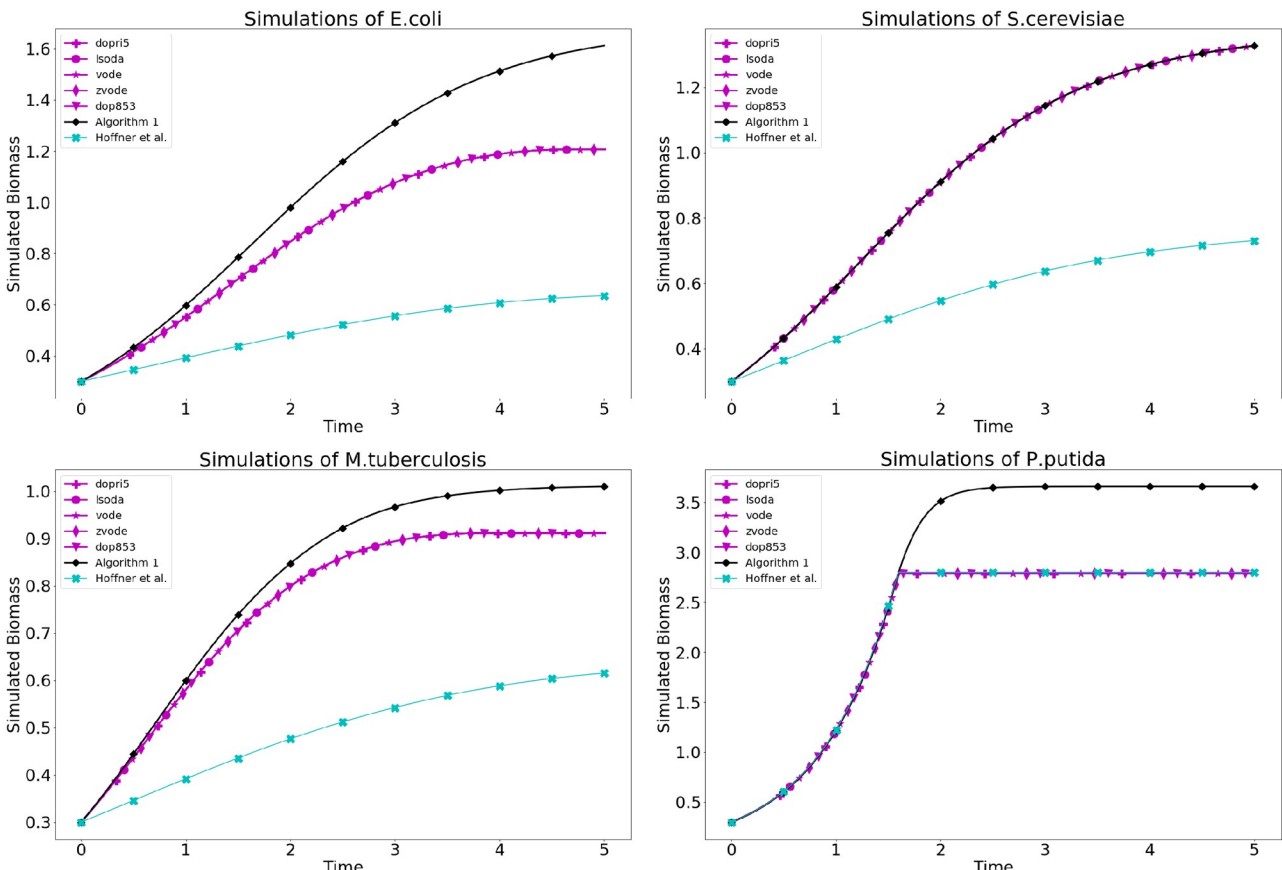

**Fig 3. Simulations of *E.coli*, *S.cerevisae*, *M.tuberculosis* and *P.putida* using Algorithm 1, direct solvers, and the method of Höffner et al.** In simulations of *E.coli M.tuberculosis*, there is discrepancy early in the simulation. In contrast, simulations of *P.putida* agree up to the point that an ODE solver fails.

*tuberculosis*, error seems to begin accumulating immediately. Finally, in the case of *P.putida*, the simulations agree well up to some time-point at which the direct method fails and either quits entirely (as in the case of the *dopri5* solver which returns small error) or continues at a constant value.

We note that discrepancies in dynamic FBA simulation may not always be due to numerical error, but instead due to non-uniqueness in optimal flux solutions. Our method provides a strategy for choosing between non-unique representations (in the form of a basis) of a single optimal flux solution. The method of Höffner et al. [40] provides a lexicographic strategy for choosing between non-unique optimal flux solutions based on biological, rather than mathematical, considerations. We note that for complete reproducibility, our method should be integrated with some biologically based strategy for choosing between non-unique optima.

## Examples & applications

There has been a recent surge in interest in modeling microbial communities using genome-scale metabolic models, much of which has focused on equilibrium methods [4, 21, 22, 26, 51]. In order to capture transient behavior and dynamic responses to stimuli, dynamic FBA has also been applied to microbial communities [24, 34, 52]. However, community dynamic FBA invariably leads to a large dynamical system with a high-dimensional parameter space, often

with little to no knowledge of parameter values. Any parameter fitting therefore requires repeated numerical simulation of the system. Existing tools to do this are built around a direct simulation approach, requiring many linear program solutions. By drastically reducing the number of optimizations required for numerical simulation, our approach offers the promise of efficient numerical simulation of dynamic FBA which will make parameter fitting more tractable, and may even allow conclusions without well-fit parameters.

Below, we demonstrate that the problem of parameter fitting is an important one by showing that experimental outcome in even small communities is sensitive to changes in kinetic parameters. Precisely, the kinetic parameters governing the uptake rate of nutrients (i.e., the parameters of the functions $c_i^2$ in Eq (4)) have a profound effect on species competition.

Next, we show how repeated simulation with randomly sampled parameters can provide some insight into community structure even without a well-fit set of nutrient uptake parameters. These examples demonstrate the importance of efficient dynamic FBA to microbial community modeling.

## Prediction dependence on nutrient uptake

The set of unknown functions $c_i^2(\boldsymbol{y})$ in Eq (4) present a profound problem for dynamic FBA simulation. If the behavior of the system is sensitive to the functions chosen and parameters of those functions, a single simulation will be of little use in drawing biological conclusion. In order to demonstrate that such a sensitivity exists, we repeatedly simulated the same simple community with different randomly drawn parameters. While a more realistic choice of function may be saturating or sigmoidal (as with Hill or Michaelis-Menten kinetics), for the following experiment we take these functions to be linear:

$$c_{ij}^2(\boldsymbol{y}) = \kappa_{ij} y_j, \tag{21}$$

meaning that the maximum uptake rate of nutrient $y_j$ by organism $x_i$ is proportional to the concentration of $y_j$. This choice minimizes the number of parameters that must be chosen for our analysis of parameter sensitivity, and is in line with an assumption of simple mass action kinetics [53, 54].

The choice of $\kappa_{ij}$ may have a profound effect on the outcome of a community simulation, as it represents how well an organism can sequester a resource when this will optimize the organism's growth. In order study this effect in a small community, we sampled a three-species community model with $\kappa_{ij} \in (0, 1)$ chosen uniformly at random. We used models for *E.coli*, *S. cerevisiae* and *M.tuberculosis* downloaded from the BiGG model database [28].

We simulated with no dilution of metabolites or microbes, and no replenishment of nutrients. In every simulation, some critical metabolite was eventually depleted and the organisms stopped growing. We recorded the simulated final biomass of each organism from each simulation, and the results are shown in Fig 4.

## Community growth effects

As we saw in previous section, community growth outcomes depend on the choice of nutrient uptake rates $\kappa_{ij}$. Using Algorithm 1, we can perform Monte-Carlo sampling in order to understand the possible effects on some microorganism of growing in some community. To do this, we randomly sample the set of uptake rates $\kappa_{ij}$ and run simulations of various communities for the chosen uptake rates. Then, the correlation between communities of final simulated biomass of some organism can be interpreted as the effect of the community on the growth of that organism. A correlation less than 1 between growth of an organism in different communities indicates that the community is having some effect. To see the direction of this effect, we can fit a simple

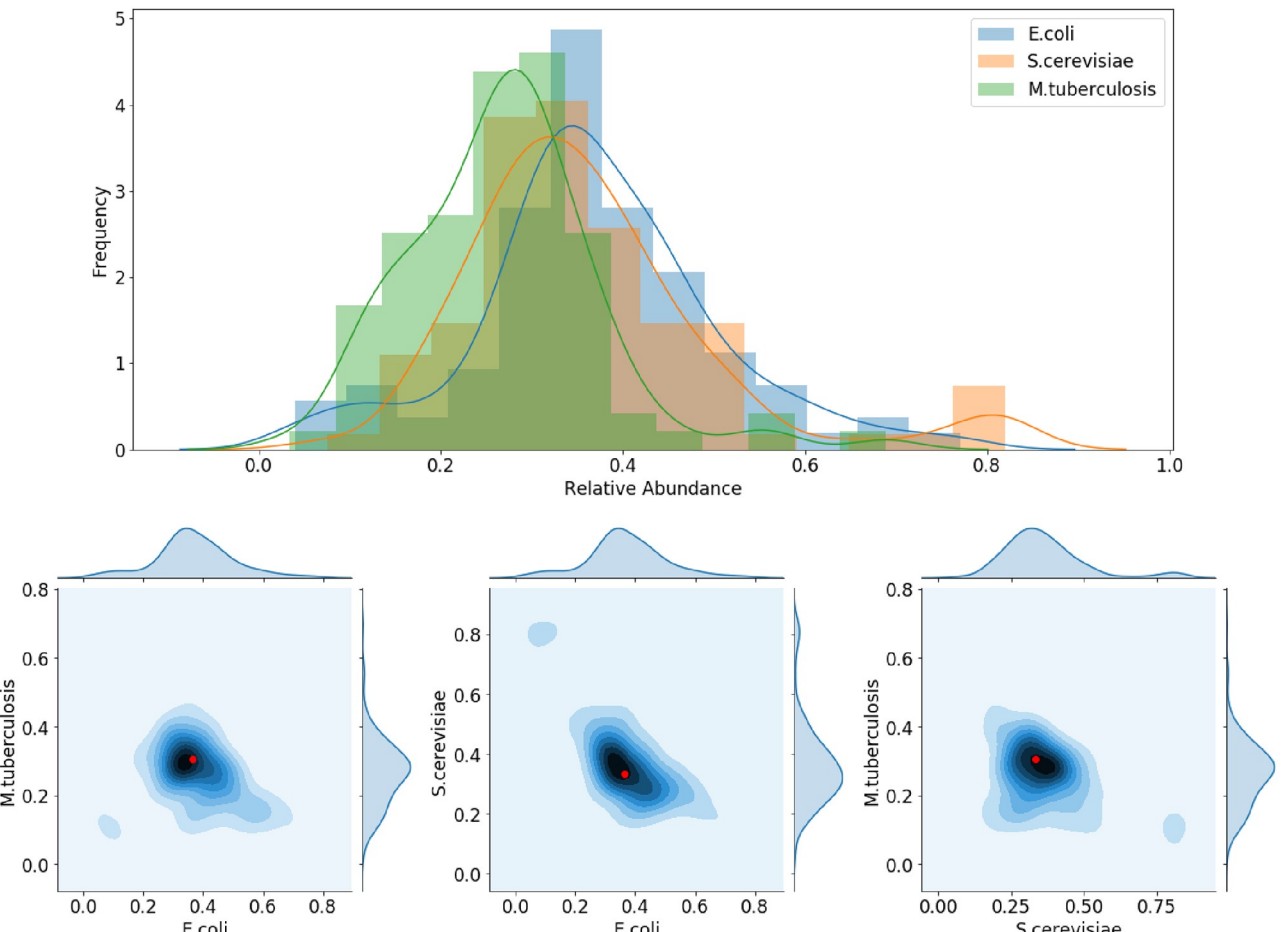

**Fig 4.** (Top) Histogram of the final simulated biomass of each of *E.coli*, *S.cerevisiae* and *M.tuberculosis* from 95 simulations, each with different metabolite uptake rates $\kappa_{ij}$. (Bottom) Pair-wise comparison of the final simulated biomass densities using a kernel density estimation. In red is the result of uniform uptake rates $\kappa_{ij} = 1$ for all $i, j$.

linear regression model (best fit line) to the final simulated biomasses. Then, the slope of this line tells us if the organism benefits or is harmed by being in one community over another.

We again simulated *E.coli*, *S.cerevisiae* and *M.tuberculosis* downloaded from the BiGG model database [28]. Simulations were run with the M9 medium described in S1 Table, with no replenishment of resources.

Each organism grew to a larger final simulated biomass when alone compared to when in a trio with the other two, which is unsurprising given the finite resources. This difference was the least pronounced for *S.cerevisiae*, suggesting that this organism is the least negatively effected by the competition. However, this can be seen as only a preliminary observation without better estimates of uptake parameters. Best-fit lines are shown in Fig 5. Efficient dynamic FBA allows repeated simulation with randomly sampled parameters, which gives an indication of likely behavior even without accurate parameter fitting.

## Conclusion

Understanding, predicting, and manipulating the make-up of microbial communities requires understanding a complex dynamic process. Genome-scale metabolic models provide an

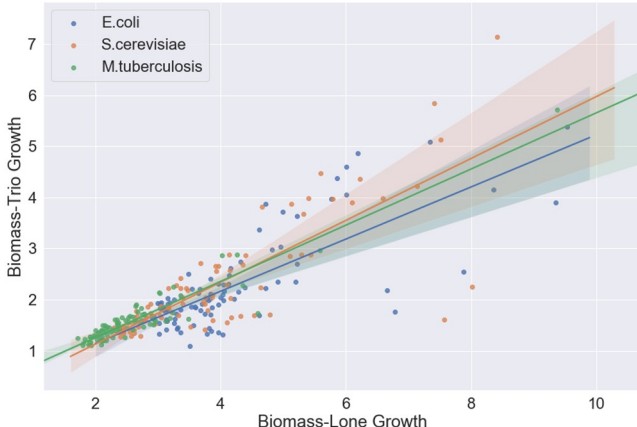

**Fig 5. Final simulated biomass of *E.coli*, *S.cerevisiae* and *M.tuberculosis* when grown alone or in pairs, for randomly sampled modeled parameters.** Best fit lines indicate the average effect of the community on an organism's growth.

approximation to this process through the quasi-steady state assumption which leads to dynamic flux balance analysis. However, this system is large and hard to simulate numerically, let alone analyze for qualitative behaviors. As a first step towards a thorough analysis of community of organisms modeled with dynamic FBA, an efficient method of numerical simulation would provide an essential tool. However, modern tools for simulating dynamic FBA rely on repeatedly solving an optimization problem at every time step [24, 31, 35–38].

Dynamic FBA simulation can be improved by considering the structure of these linear programs so that many fewer optimizations are required. As of now, the algorithm of Höffner et al. [40] is the only published method which takes advantage of this observation. However, that method does not account for the degeneracy of solutions to the relevant linear programs, meaning that it can choose a solution that cannot be carried forward in time. We present a method that chooses a basis for forward simulation. In contrast to the method of Höffner et al., we choose this basis in such a way that increases the likelihood that this forward simulation is actually possible.

Efficient dynamic FBA will allow better parameter fitting to time-longitudinal data. Furthermore, it allows for a search of parameter space which can help predict likely model outcomes or learn maps from parameter values to model outcomes.

## Supporting information

**S1 Text. Proof of Lemma 1.** Proof of the main lemma of the paper.
(PDF)

**S1 Table. M9 medium File.** Defines an M9 minimal medium as adapted from Monk et al. [55].
(CSV)

**S2 Table. List of models used.** Provides name, ID, and URL for the four models used in analysis of the method.
(CSV)

## Author Contributions

**Conceptualization:** James D. Brunner, Nicholas Chia.

**Data curation:** James D. Brunner, Nicholas Chia.

**Formal analysis:** James D. Brunner.

**Funding acquisition:** Nicholas Chia.

**Investigation:** James D. Brunner, Nicholas Chia.

**Methodology:** James D. Brunner, Nicholas Chia.

**Project administration:** James D. Brunner, Nicholas Chia.

**Resources:** James D. Brunner, Nicholas Chia.

**Software:** James D. Brunner, Nicholas Chia.

**Supervision:** Nicholas Chia.

**Validation:** James D. Brunner, Nicholas Chia.

**Visualization:** James D. Brunner.

**Writing – original draft:** James D. Brunner.

**Writing – review & editing:** James D. Brunner, Nicholas Chia.

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
