## [Decision Letter · Decision Letter 0]

28 May 2020

Dear Dr. Brunner,

Thank you very much for submitting your manuscript "Minimizing the number of optimizations for efficient community dynamic flux balance analysis." for consideration at PLOS Computational Biology.

As with all papers reviewed by the journal, your manuscript was reviewed by members of the editorial board and by several independent reviewers. In light of the reviews (below this email), we would like to invite the resubmission of a significantly-revised version that takes into account the reviewers' comments.

We cannot make any decision about publication until we have seen the revised manuscript and your response to the reviewers' comments. Your revised manuscript is also likely to be sent to reviewers for further evaluation.

Sincerely,

Pedro Mendes, PhD

Associate Editor

PLOS Computational Biology

Stefano Allesina

Deputy Editor

PLOS Computational Biology

Reviewer's Responses to Questions

**Comments to the Authors:**

Reviewer #1: The authors propose a method for efficient dynamic flux balance analysis (DFBA) of community metabolic models that reduces the number of LP solutions as compared to brute-force methods based on solving the LP at each time step. The method is applied to an 8-species community to demonstrate its usefulness. While the development of numerical methods for community DFBA is an important problem, the manuscript has several shortcomings that need to be addressed before publication can be recommended.

1. First and foremost, the authors fail to explain why their method is superior to or even different from the DFBA method proposed by the Barton group at MIT [33, 34]. The two methods seem very similar as suggested by the authors: "Hoeffner et al, [34], used this observation to introduce a variable

step-size method for dynamic FBA. In that method a basic index set is chosen by adding biological constraints to the optimization problem hierarchically until a unique optimal solution is found. The challenge of such an approach is in choosing the basis for the optimal solution, as the optimal basis is not guaranteed to be unique. We describe how to choose a basis from among the possibilities provided from an FBA solution which is most likely to remain optimal as simulation proceeds forward. We therefore prioritize reducing the number of times the linear program must be solved." So the claim innovation here is more rational choice of the optimal basis? This point is critical and not clearly explained.

2. While the authors are correct that the COBRA toolbox offers a simple and inefficient Euler-based method (e.g. [30]), I would hardly consider this method to be state-of-the-art. If the authors would like to claim superiority, then they should compare their method to the DFBA method proposed by the Barton group [33, 34].

3. When comparing methods, the authors state that the Euler-type DFBA method required a very short time step to ensure non-negative metabolite concentrations. Perhaps a more efficient implementation of the Euler method would be to utilize an ODE solution method that ensure non-negativity of the dependent variables, such as the ODE solvers available in MATLAB. Did the authors consider this option, which would allow a fairer comparison?

4. The application to the 8-species community is a reasonable test of the method, but some biological context also would be useful. For example, why focus on uracil as a crossfed metabolite? Why assume linear linear uptake kinetics rather than the usual Michael-Menten kinetics?

5. A computation time of over 15 minutes for a 1 hour simulation (see Fig. 5) does not seem particularly fast to me. Please compare computation times with a state-of-the-art DFBA code.

6. Several of the plots lack axes labels (e.g. Fig. 6, 7 and 9). Please correct.

Reviewer #2: In this manuscript, Brunner and Chia have proposed an algorithm to accelerate the simulation of time-course behavior of microbial communities. The existing dFBA algorithm was improved upon by implementing a solution-hopping strategy in order to reduce the number of times the FBA problem must be solved at each time step. The authors have shown that this approach remains scalable for communities containing up to 8 members when compared to the classic dFBA algorithm. Overall, this speed-up is significant and when properly benchmarked against existing state-of-the-art algorithms, has the potential to be an important advancement in the field of microbial community modeling.

Major comments:

1. The authors have only compared their algorithm against the classical dFBA algorithm. However, in recent years, several meaningful algorithms have been developed such as d-OptCom (Zomorrodi and Islam et al, ACS Synthetic Biology, 2014), COMETS (Harcombe et al, Cell Rep., 2014), and SteadyCom (Chan et al, PLoS Comput Biol, 2017). The authors must benchmark their method in terms of both time and performance with the above methods.

2. The proposed method assumes that maximization of total biomass is the community objective. A consequence of this is that only the fastest growing organism will show any growth over time and cause the community to become a monoculture. The authors must comment on how their method safeguards against such outcomes.

3. The classical dFBA algorithm consists of a numerical ODE integration wrapped around an instantaneous FBA problem. It is the requirement of 10,000 integration steps that leads to the high computational cost. As a result of this, the reported “85% reduction in computation time” may be inflated. It would be more meaningful to compare the two algorithms using an adaptive time-step based integration such as adaptive Runge-Kutta or Richardson’s extrapolation, or a faster ODE solver such as CVODE from the SUNDIALS package.

4. Because the forward simulation method approximates the optimal FBA solution at multiple time steps, the authors must also compare the accuracy of their simulations with that of classical dFBA and comment on the accuracy vs. speed trade-off of their proposed algorithm.

5. Metabolic fluxes have a large dynamic range of values spanning up to three orders of magnitude. This can often lead to numerical instability issues where concentrations assume negative values. The authors must justify why 10,000 time-steps are sufficient to bypass this issue.

6. The authors have stated that re-solving is necessary when the solution becomes infeasible. However, metabolic networks are carefully constructed such that, as long as concentration of species in the extracellular media remain non-negative, the FBA problem will never be infeasible. The authors must clarify what leads to infeasibility in their FBA problem. Could this be an artefact of the approximation introduced in this algorithm to bypass re-solving the FBA problem?

7. The authors must also include a discussion on the new analyses enabled by the accelerated simulation of microbial communities. In particular, what previously difficult to answer questions can now be addressed by the community due to the advancement afforded by this proposed method.

Minor Comments:

1. The authors must clarify that their method improves upon the Static Optimization Approach of dFBA.

2. In the formulation shown in Equation (4), the quantity “c” is used prior to its definition. Furthermore, the function c(y) has not been defined in this manuscript.

3. It is recommended that different subscripts be used to denote microbial species, metabolites, and reactions for clarity of presentation. The same subscript “i” has been used to represent all of these quantities.

4. In equations (6), (7), and (8), it is unclear why all fluxes are represented in terms of intracellular fluxes only. Generally, the kernel of the stoichiometry matrix is represented in terms of all but one extracellular exchange fluxes, and the intracellular fluxes causing a rank-deficiency in the stoichiometry matrix.

Reviewer #3: The authors present a new algorithm (surfinFBA) for computing dynamic FBA simulations which does not require solving the linear problem at every time step, which greatly improves the computation times. Although the method looks promising, the quality of the manuscript is not so great and, in my point of view, needs substantial restructuring.

Major issues:

* The main emphasis of the paper is put on the mathematical details of deriving the algorithm, which would be interesting in a mathematical / computer science journal, but is not so interesting to the target audience of this journal. I would suggest moving these details to a supplementary text, and focusing the main text on the simulation results and benchmark of the new method.

* The authors present the classic and new algorithm as Figures in the manuscript, and throughout the text often refer to running these algorithms as implementing / simulating Fig.1 and Fig. 3. I find this inappropriate and very confusing. The two algorithms should be presented as methods in the Methods section and referred to by their names, not by figure numbers.

* The authors cite the first dFBA method by Varma and Palsson (1994), but they should also mention the first implementation of dFBA for multi-species simulation, the DyMMM method by Zhuang and co-workers (2011).

* There are also more recent multi-species dFBA methods like MCM (Louca and Doebeli, 2015) and microbialSim (Popp and Centler, bioRxiv, 2019). The authors should also compare the advantages / disadvantages of their method with these more recent methods. For instance, in their pre-print, Popp and Centler show that microbialSim as able to simulate a 773-species community, a much larger size than the 8-species community simulated with surfinFBA.

* The authors should better classify how they setup the simulation conditions. In the paper they mention they use the "COBRApy default simulated medium". This does not make sense. COBRApy is just a simulation library, it makes no assumptions regarding the medium. The medium is something the user must decide upon when parameterizing a model.

Minor issues:

* The authors mention they downloaded the models from the PATRIC database, which as far as I know is a genome database, not a model database, and I could not find how to download an SBML model from PATRIC. The models provided in the github repository are JSON files with modelSEED annotations, so I supposed they were built with modelSEED. This should be better clarified.

Reviewer #4: We have read “Minimizing he number of optimizations for efficient community dynamic flux balance analysis” by Brunner and Chia. The authors have developed a method named surfinFBA for performing dynamic flux balance analysis calculations for in-silico communities of organisms that reduces the number of optimization problems that must be solved. This approach leads to an 85% reduction in compute time required relative to the standard method implemented in the COBRA toolbox (a package frequently used by the genome-scale modelling community). We applaud the authors use of github to disseminate the method in a usable format. Our interest in this manuscript is only tempered by a lack of discussion, and insufficient benchmarking relative to the classical method performed by the authors.

Major comments

1. The author demonstrates that surfinFBA provides a 85% speed increase in dynamic FBA simulations relative to the standard method. Supplementary File 1 compares surfinFBA to the classic method implemented in the COBRA toolbox. We appreciate seeing the time differences however there is no comparison of results between the two implementations. The author must add an analysis demonstrating the results are equivalent between the two tools, otherwise the time differences is less meaningful. Adding computed dFBA profiles from both the classic method and the novel method to the supplementary file for comparison would be a first step.

2. A key value of FBA and dFBA simulations is their mechanistic nature enabling interpretation and contextualization of results. Thus, we were disappointed in the statement on line 436 where the authors mention that P. veronii had “some competitive advantage” in the default media. What media, specifically was used? Models usually come pre-loaded with a default media, but the COBRA toolbox itself does not (this should be corrected in the manuscript). The author should utilize the simulation results to trace the media component providing this competitive advantage to P. veronii. What aspect of the metabolic network for P. veronnii provides this competitive advantage? A deeper dive into some of the results presented here would provide more value to the manuscript.

3. The manuscript ends rather abruptly, and we were disappointed in the lack of discussion section. Now that dFBA simulations can be run more efficiently what applications do the authors envision? The applications presented at the end are informative but additional discussion of their value would be illuminating.

4. Please provide details of all simulations performed. The stoichiometric matrices and flux bounds used should be provided in an excel sheet

Minor comments

1. Be sure to distinguish between the python and matlab implementations of the cobra toolbox – I believe all references are to the python implementation but please make this explicit. Also reference the version used for benchmarking.

2. Be sure to label axes in all figures – e.g. 6 and 7.

**Have all data underlying the figures and results presented in the manuscript been provided?**

Reviewer #1: Yes

Reviewer #2: Yes

Reviewer #3: No: Numerical data underlying figures do not seem to be provided according to the data availability policy of the journal.

Reviewer #4: No: Simulation parameters used to generate the results should be shared. The stoichiometric matrices and flux bounds used should be provided in an excel sheet

PLOS authors have the option to publish the peer review history of their article (what does this mean?). If published, this will include your full peer review and any attached files.

Reviewer #1: No

Reviewer #2: No

Reviewer #3: No

Reviewer #4: No
---

## [Decision Letter · Decision Letter 1]

21 Aug 2020

Dear Dr. Brunner,

We are pleased to inform you that your manuscript 'Minimizing the number of optimizations for efficient community dynamic flux balance analysis.' has been provisionally accepted for publication in PLOS Computational Biology.

Best regards,

Pedro Mendes, PhD

Associate Editor

PLOS Computational Biology

Stefano Allesina

Deputy Editor

PLOS Computational Biology

Reviewer's Responses to Questions

**Comments to the Authors:**

Reviewer #1: The authors undertook some serious revisions to address my previous comments and concerns. Therefore, I am satisfied with the revision.

Reviewer #2: The authors have sufficiently addressed our raised concerns.

Reviewer #4: The authors have performed a significant amount of work to address all of our comments and suggestions

We encourage them to comb through the manuscript one last time for grammatical errors that were introduced in the revision

We noticed a few, e.g. line 601 "by show that"

Besides these very minor issues we feel the the manuscript has been greatly improved through the revision.

**Have all data underlying the figures and results presented in the manuscript been provided?**

Reviewer #1: Yes

Reviewer #2: Yes

Reviewer #4: Yes

PLOS authors have the option to publish the peer review history of their article (what does this mean?). If published, this will include your full peer review and any attached files.

Reviewer #1: No

Reviewer #2: No

Reviewer #4: No

---

## [Editor Report · Acceptance letter]

24 Sep 2020

PCOMPBIOL-D-20-00387R1 

Minimizing the number of optimizations for efficient community dynamic flux balance analysis.

Dear Dr Brunner,

I am pleased to inform you that your manuscript has been formally accepted for publication in PLOS Computational Biology. Your manuscript is now with our production department and you will be notified of the publication date in due course.

With kind regards,

Sarah Hammond
